# A Review of Nutritional Water Productivity (NWP) in Agriculture: Why It Is Promoted and How It Is Assessed? †

Katrin Drastig [1,*] , Ranvir Singh [2], Fiorina-Marie Telesca [1], Sofia Zanella Carra [1] and Jasper Jordan [1]

1   Leibniz Institute for Agricultural Engineering and Bioeconomy, Max-Eyth-Allee 100,
    14469 Potsdam, Germany; fiorinatelesca@gmail.com (F.-M.T.); sofiazcarra@gmail.com (S.Z.C.);
    jasperjordan@web.de (J.J.)
2   Environmental Sciences Group, Massey University, Palmerston North 4442, New Zealand;
    r.singh@massey.ac.nz
*   Correspondence: kdrastig@atb-potsdam.de; Tel.: +49-331-5699-218
†   OECD disclaimer: The opinions expressed and arguments employed in this publication are the sole
    responsibility of the authors and do not necessarily reflect those of the OECD or of the governments of its
    Member countries.

**Abstract:** Assessment of nutritional water productivity (*NWP*) combines a metric of crop or livestock production per unit water consumed and human nutritional value of the food produced. As such, it can rationalize the use of scarce water for a portfolio of crop and livestock production systems that jointly match human nutritional needs and reduce water scarcity impacts. However, a comprehensive search and review of 40 *NWP* studies highlighted that current *NWP* studies vary widely in terms of their methodological approaches, the data and tools used and the water flows and nutrient content accounted for. Most of the studies accounted for evapotranspiration stemming from precipitation and technical water, and/or inclusion of the withdrawn technical water. Water scarcity was only addressed in four studies. The reported *NWP* values also varied for accounting of macro- (energy, protein, fat and carbohydrates) and micro-nutrient (minerals and vitamins) content. The methodological differences, however, severely limit the informative value of reported *NWP* values. A multidisciplinary research effort is required to further develop standardized metrics for *NWP*, including its local environmental water scarcity impacts. A robust *NWP* analysis framework in agriculture should focus on the integration of assessments of *NWP* and water scarcity impact (*WSI*), and development of more field measurements and locally calibrated and validated agrohydrological and farm production models to quantify reliable *NWP* values and their associated *WSI* of agriculture production systems worldwide.

**Keywords:** nutrient water productivity; nutrition security; food groups; water demand; water consumption; water withdrawal; technical water; evapotranspiration; macronutrients; micronutrients





## 1. Introduction

Growing population and changing consumption patterns are putting increasing pressure on limited land and water resources for provision of water and food securities, especially in developing countries. For instance, the global water withdrawal has increased by 630% during 1900–2010 [1]. Climate change is projected to further affect water availability, as extreme weather events and altering rainfall patterns will pose immense water challenges on the agricultural sector [2]. Given that about 92% of all fresh water worldwide is used by agriculture, finding ways to use less water and use it more productively is highly important in dealing with the growing problem of water scarcity and increasing demands of water for agriculture [3].

### 1.1. Current Metrics Commonly Used for Assessing the Productivity of Water Use in Agriculture

Various approaches have been developed to assess water use agriculture at different scales. Agricultural scientists often work on the crop or field scale to express the relation of

water use to output generation, in terms of water efficiency or productivity. The virtual water concept has been developed from an economical perspective. It provides a basis for estimating the water demand for agricultural products and the derivation of the water footprint of nations from their agricultural production and food trade. Water use impact assessments are primarily carried out on a catchment scale, i.e., covering the extent of land sharing a common drainage basin and the scale at which agriculture impacts water scarcity.

A consistent approach to water use metrics is required for a robust assessment of how different crops and livestock production systems and their management practices influence water use and its productivity in agriculture, and accordingly inform changes to reduce water use while increase food production. The existing water use assessment approaches fall into three main categories [4] as follows:

- Water productivity (*WP*), e.g., [5–7];
- (Volumetric) water footprint (*WFPa*), e.g., [8] based on the concept of virtual water [9];
- Water scarcity footprint (*WFPb*), based on the Life cycle assessment (LCA)-based/ISO 14046:2014 [10].

Current water use assessment metrices link input and output variables differently, resulting in variable interpretations of water use data to help inform productivity and sustainability of water use in agriculture. Water productivity (*WP*) describes an output to input relationship: the ratio of output [e.g., *kgFM*, kg fresh matter potatoes] to productive water input [e.g., m$^3$ transpiration, *T* or evapotranspiration, *ET*], e.g., [11–13].

The water component could include both direct water use (e.g., use of precipitation and/or irrigation on a farm) and indirect water use (e.g., purchased feed on a farm) [14,15]. The distinction between direct and indirect water use is based on operational use (direct water consumption) and supply chain use (indirect water consumption). The consumption of precipitation and irrigation is assessed by crop water use modelling as evapotranspiration stemming from precipitation and technical water (surface water and groundwater) used for irrigation on the farm. Prochnow et al. [16] noted that the question of how to treat soil evaporation and plant transpiration from infiltrated precipitation is highly controversial among scientists involved in estimating water use in agricultural production. Hydrologists balance water inflows and outflows at different levels such as fields or watersheds and must include evapotranspiration. From an agricultural perspective, many scientists only consider evapotranspiration or transpiration as inputs for assessing water efficiency at different scales.

The *WP* method [5–7]; is mainly applied to assess farming measures to improve productive use of precipitation and technical water (surface and groundwater used for irrigation, processing and stock drinking and services) in livestock and crop production. Developing a consistent and coherent approach for assessing water productivity could lead to a benchmark for the water productivity values of different production systems in specific regions [4]. Prochnow et al. [16] proposed that water productivity should be calculated on different output bases (mass, monetary, energy), and that the water productivity of (feed) crops, the level of water utilization and the specific inflow of technical water are reported and assessed separately. The water footprint (*WFPa*) describes an input to output relationship: the ratio of water input [m$^3$, evapotranspiration *ET*] to output [kilo of fresh matter, *kgFM*] of a production system [8]. It also includes direct and indirect water uses, accounting for water consumption and pollution throughout the production chain [8]. The water footprint (*WFPa*) of a crop production (e.g., m$^3$ *ET* per kg *FM*) is an inverse of the crop *WP* (e.g., kg *FM* per m$^3$ *ET*). However, the *WFPa* could be quantified as a metric of water consumed in both the production and consumption of agricultural products [8].

The quantification of process water footprints offers a building block to quantify water footprints of a product, producer, industry sector, consumer or community (a city, province or nation) [8]. However, the volumetric quantification of *WFPa* does not account for the water scarcity impacts associated with the water consumed in a geographical location. The water scarcity footprint (*WFPb*) further contextualizes the water consumed with respect to the water availability to assess environmental impacts of water use in agriculture based

on ISO 14046:2014 [10]. Xu et al. [1] mentioned water scarcity as an indicator for water use in relation to water security. Water scarcity indicates the shortage of renewable fresh water compared to the amount of fresh water demanded in a certain area. The life cycle assessment LCA-based *WFPb* method analyses water scarcity impacts mainly on a regional scale and takes into account only technical water used [17]. This follows the argument that only technical water (surface water and groundwater) flow is relevant in contributing to local water scarcity.

The FAO Livestock Environmental Assessment Programme (LEAP) recently organized a Water Technical Advisory Group (Water TAG) to review different water use assessment approaches, methods and tools relevant to livestock production systems and supply chains [18]. The FAO LEAP Water TAG recommended a consistent combination of water productivity and water scarcity footprint metrics as a harmonized approach for assessing livestock water use in terms of potential water productivity improvements and minimizing potential water scarcity impacts [18,19].

However, as Fereres and Soriano [20] stated, the water use metrics based on simple ratios, such as *WP* or *WFP* do not offer enough information to guide policies or to make meaningful comparisons of different food production systems to ensure food nutrition security. For example, comparison of 1 *kgFM* of a crop (e.g., wheat) does not offer same nutrient value compared to 1 *kgFM* of another crop (e.g., sugarcane). This calls for using an improved metric to measure nutritional water productivity in agriculture.

*1.2. Nutritional Water Productivity as a Metric Linking Water, Agriculture, Nutrition, and Human Health Outcomes*

The issues we face regarding global hunger, and the ways to tackle them, are often discussed regarding food security. However, this term disregards an essential factor, namely nutrients. Food security promotes an increase in the quantity of foods produced, overlooking the importance of the quality of food produced [21].

In recent years, lack of nutrition has gained more traction as experts have recognized that food security cannot be achieved without nutrition security [22]. Nutrition security has become an increasingly more established term used alongside with the term food security, as together they highlight the need for both food and health requirements [22]. To tackle both issues of increasing water scarcity and lacking food and nutrition security, there is a need to recommend the cultivation of nutrient-dense crops that are adapted to water-limited conditions [23]. Recalling the need for improved metrics to measure and compare the effects of different agricultural practices on water productivity, Wenhold et al. [24] suggested a shift in how we measure water productivity in agriculture from 'more crop per drop' to 'more nutrition per drop'.

Nutritional water productivity (*NWP*) has been proposed as a water use metric that can be used to measure water productivity in terms of nutrient values produced per unit of water used, and thus make recommendations on how to grow more nutritious crops while being more productive with water use [23]. *NWP* is suggested as a robust measure of the quantity and quality of food produced per unit of water used in agriculture [25]. Increasing *NWP* is promoted as a productive and sustainable method in agriculture because it optimizes the use of limited water resources while maximizing production of food nutrition and improving food and nutrition security and human health outcomes. This is especially important in regions where water resources are scarce and demands for food nutrition are high, especially in many Asian and African countries.

Assessing *NWP* typically involves measuring the quantity of water used in agriculture production and the amount and quality 'nutrient content' of produced [25]. The formula for the calculation of *NWP* was first introduced by Renault and Wallender [25] as follows:

$$NWP = (Y/ETa) \times NP \qquad (1)$$

where, *NWP* is the nutritional water productivity [nutrition unit/m$^3$ of water], *Y* is the actual harvested crop yield [kg/ha], *ETa* is the actual evapotranspiration [m$^3$/ha], and *NP* is

the nutrition content per kg of the crop/product yield [nutrition unit/kg]. Nutrient content can be macronutrients (e.g., energy, protein, fat and carbohydrates) and micronutrients (e.g., minerals and vitamins: Vitamin A, Iron, Zinc and calcium), carbohydrates, fibre, and starch. However, limited studies have applied and discussed *NWP* as a metric for water use assessment in agriculture so far. This paper presents a comprehensive review of the current use of *NWP* in the literature, including its definition, analysis objectives and any potential limitations and recommendations for *NWP* in agriculture. We assess potential of *NWP* as a suitable metric for measuring water use in agriculture in relation to human nutrition security, and in turn as a valuable tool for addressing water scarcity and nutrition security. We firstly constructed an overview on the current state of development and application of the novel concept of *NWP*, and then evaluated the usefulness of *NWP* as a metric for agricultural water use assessment to address current issues of water scarcity and nutrition security. The literature review aimed to identify potential limitations and offer recommendations for future research and *NWP* applications to inform the development of productive and sustainable water use in agricultural production systems.

## 2. Material and Methods

To conduct a systematic literature search and review, we developed and used eight strings of keywords to search in two databases: 'Web of Science' and 'Google Scholar' (Figure 1). Our literature search, using combinations of keywords and two databases, aimed to find both scientific and technical literature focused on *NWP* directly and indirectly, given that some publications might use a similar concept with a different term used for the definition of nutritional water productivity in agriculture.

**Subject area:** Water use in agriculture, environmental assessment, nutrient water productivity
**Language:** English, Portugues
**Resulting timespan:** 01.2000 – 05.2023

| Criterias | Selection of Database | Key words | Broad inclusion criteria | Publications from other sources | Publications in total |
|---|---|---|---|---|---|
| | Web of Science Google Scholar | "nutrient water productivity" "nutrition water footprint" "nutrition a water footprint" "water nutrient density" "water nutritional density" "nutrition density of crops" "nutritional density of crops" | Whether the publication presents a calculation of nutrient water productivity | Based on experts knowledge and upon assessment publications | |
| | | Collection of articles Web of Science: 17 Google scholar: 24 6 overlaps | First stage evaluation by scanning the title and article | | |
| Number of articles | | 35 publications | Leaving 33 publications | 7 publications | 40 publications in total |

**Figure 1.** An overview of the literature search engines, keywords used and number of publications identified.

In the literature search, the articles were selected if the search keywords (Figure 1) appeared in the title, the keywords list, or the abstract using Web of Science. On Google scholar, the articles were selected if the search keywords appeared in the title only, as it was impossible to choose the keywords appearing in the keywords list or the abstract. The results with the search keywords used appearing anywhere in the text were too many, with the keywords mostly only appearing in the general text descriptions, and thus not relevant for the focused literature review here. English publications and one publication in Portuguese language were considered. There were no limitations set on the literature

timeline, considering that the concept of *NWP* is a relatively new and emerging area of scientific research and analysis. Therefore, not many articles related to *NWP* have been published and there was no need to further limit the search. These publications were then analysed against broad inclusion criteria following Jia et al. [26].

Based on expert knowledge and upon further assessment, a total of seven publications [3,24,27–31] from other sources were also added to the list of relevant publications for the literature review.

Our systematic literature search finally resulted in the 40 publications that were analysed in this literature review. The publication from Brooks and Grauenhorst [32] cited by Wenhold et al. [24] and Mabhaudhi et al. [21] was not available and for this reason could not be included into the review. The collated literature review database included 25 scientific journal papers, 9 graduation theses (gray articles), 5 technical reports and 1 conference paper. The selected 40 publications were analysed according to their published values of *NWP* for single crops and livestock products ordered into different food groups and primary derivatives thereof. The chosen food groups and feed oriented towards the classification in Destatis [33–35], EFSA [36], FAO/INFOODS [37] and USDA [38]. The collated studies were reviewed in terms of their assessment objectives, methodological approaches, production system and scale of analysis, data sources and modelling tools used and the different water flow- and nutrient content analysed, offering insights into 'state-of-the-art' *NWP* analysis in agriculture.

## 3. Results and Discussion

### 3.1. Key Objectives of NWP Studies

The key objectives the of *NWP* studies, as expected, were focused on "benchmarking/reference values for *NWP* values", "assessment of farming measures", "assess/recommend/improve diets", "improved understanding of water use in different production and consumption systems" and "method development" over a range of the main food groups and feed produced in agriculture (Figure 2).

In terms of food groups, 9 studies included animal products (meat, milk and eggs); 5 studies included feed or feed crops; 9 studies included nuts and/or fruits; and 38 studies included crops (Figure 2). The majority of the studies calculated the *NWP* of single crops (13 studies) or multiple crops (25 studies), while 11 studies analysed different diets.

A total of 13 studies have focused on separately analysing of NWP of single crops (Table 1).

**Table 1.** Focus of existing studies analysing nutritional water productivity of single crops ordered into different food groups and primary derivatives thereof. The chosen food groups and feed oriented towards the classification in Destatis] [33–35], EFSA [36], FAO/INFOODS [37] and USDA [38].

| Food Group | Crop | Publication |
|---|---|---|
| cereal grains and similar and primary derivatives thereof | sorghum, rice | Hadebe et al. [39,40], Kapuria [41] |
| starchy roots and tubers | potatoes | Dladla [23] |
| | sweet potatoes, | Mulovhedzi et al. [42] |
| | taro [*Colocasia esculenta* (L.) Schott] | Nyathi [43] Mabhaudhi [44], Shelembe [45] |
| legume seeds and primary derivatives thereof | cowpea | Kanda et al. [46] |
| | sutherlandia frutescens | Masenya [47] |
| garden vegetables and primary derivatives thereof | spinacea Oralecea (fordhook giant), hot chili, tomato | Nyathi [48], Ramputla [49], Li et al. [50] |

| | Cereal grains | Garden vegetables | Legume seeds | Starchy roots and tubers | Nuts | Fruit | Meat, Milk, Eggs and products | Feed | |
|---|---|---|---|---|---|---|---|---|---|
| **Benchmarking/ reference values for NWP** | Lundqvist et al. (2021) Malmquist (2018) Masenya (2022) Sokolow et al. (2019) | Ali (2011) Nouri et al. (2020) Nyathi et al. (2018) Blas et al. (2019) Wenhold et al. (2007; 2012) — Nyathi et al. (2019b) Nyathi (2019) | Chibarabada (2018) Chimonyo et al. (2023) | Ali (2011) Wenhold et al. (2007; 2012) | Nyathi (2019) Shelembe (2020) Blas et al. (2019) | Ali (2011) Blas et al. (2019) Wenhold et al. (2007; 2012) | Ali (2011) Blas et al. (2019) Wenhold et al. (2007; 2012) | Liu et al. (2022) | |
| | Istaitih & Rahil (2018) Kunene (2020) Nyathi et al. (2016) Shelembe (2017) Liu et al. (2022) | Li et al. (2021) Nyathi (2011) | | Kunene (2020) Nouri et al. (2020) Nyathi et al. (2016) | Mabhaudhi et al. (2019) Dladla (2017) Istaitih & Rahil (2018) Mulovhedzi (2017) Nyathi et al. (2019a) | | Nouri et al. (2020) | | |
| **Assessment of farming measures** | | Nouri et al. (2020) Nyathi et al. (2018) — Shelembe (2017) Xue et al. (2021) | Kanda et al. (2020) | Shelembe (2017) | | | | Shelembe (2017) Xue et al. (2021) | |
| **Assess/ recommend/ improve diet** | Hadebe et al. (2017) Hadebe et al. (2021) Kapuria and Banerjee (2022) | Blas et al. (2019) Mirzaie-Nodoushan et al. (2020) Nyathi (2019) | Chibarabada et al. (2017) Masenya (2022) | Malmquist (2018) | Blas et al. (2019) | Blas et al. (2019) Malmquist (2018) | Blas et al. (2019) Malmquist (2018) Mirzaie-Nodoushan et al. (2020) Renault & Wallender (2000) | | |
| **Improved understanding** | | Lundqvist et al. (2021) Mabhaudhi et al. (2016) Malmquist (2018) Wenhold et al. (2007; 2012) — Blas et al. (2019) Ramputla (2019) | Mabhaudhi et al. (2016) | Lundqvist et al. (2021) Wenhold et al. (2007) | Blas et al. (2019) | Wenhold et al. (2007) Blas et al. (2019) Lundqvist et al. (2021) | Blas et al. (2019) Malmquist (2018) Wenhold et al. (2007) | | |
| | | | | Malmquist (2018) | | Malmquist (2018) Wenhold et al. (2012) | | Palhares (2013) | |
| **Method development** | Mdemu et al. (2009) Sokolow et al. (2019) Renault & Wallender (2000) | | Mdemu et al. (2009) | Sokolow et al. (2019) Renault & Wallender (2000) | Renault & Wallender (2000) | | Renault & Wallender (2000) | | |

**Figure 2.** Main objectives of the existing 40 nutrient water productivity studies for different food groups, and primary derivatives thereof, and feed. A small number of studies (*n* < 4) included oilseeds and oil fruits, fish meat, cotton, grapes, sugar plants, wild mustard, edible mushrooms, primary derivatives thereof, and/or processed foods and beverages. These were excluded for the clarity of the figure. Food groups and feed chosen in the figure oriented towards the classification in Destatis [33–35], EFSA [36],, FAO/INFOODS [37] and USDA [38]. Studies included here: Ali (2011) [51]; Blas et al. (2019) [52]; Chibarabada et al. (2017) [53]; Chibarabada (2018) [54]; Chimonyo et al. (2023) [55]; Dladla (2017) [23]; Hadebe et al. (2017) [39]; Hadebe et al. (2021) [40]; Istaitih and Rahil (2018) [27]; Kanda et al. (2020) [46]; Kapuria and Banerjee (2022) [41]; Kunene (2020) [56]; Li et al. (2021) [50]; Liu et al. (2022) [57]; Lundqvist et al. (2021) [28]; Mabhaudhi et al. (2016) [21]; Mabhaudhi et al. (2018) [44]; Malmquist (2018) [29]; Masenya (2022) [47]; Mdemu et al. (2009) [30]; Mirzaie-Nodoushan et al. (2020) [58]; Mulovhedzi (2017) [42]; Nouri et al. (2020) [3]; Nyathi et al. (2016) [59]; Nyathi et al. (2018) [60]; Nyathi (2011) [48]; Nyathi (2019) [61]; Nyathi et al. (2019a) [43]; Nyathi et al. (2019b) [62]; Palhares (2013) [63]; Ramputla (2019) [49]; Renault and Wallender (2000) [25]; Shelembe (2017) [45]; Shelembe (2020) [64]; Sokolow et al. (2019) [31]; Tompa et al. (2020) [65]; Wenhold et al. (2007) [66]; Wenhold et al. (2012) [24]; Xue et al. (2021) [67].

The main objectives of the single crop studies were largely to compare the effects of different farm management practices on the NWP of the selected crop (Figure 2). The studies by Hadebe et al. [40], Mulovhedzi [42] and Dladla [23] differentiated between assessment of different genotypes and cultivars, to examine which one is more suitable in terms of NWP. Dladla [23] also highlighted that WP and NWP were found higher under limited irrigation water applied (30% of crop evapotranspiration), as compared to the optimal water applied (100% of crop evapotranspiration), in the cultivation of sweet potatoes in KwaZulu-Natal, South Africa. Dladla [23] demonstrated the potential of NWP

as a very useful indicator of crop performance towards food and nutrition supply under different water supply conditions.

Several studies focused on assessing the *NWP* of several crops, ranging from four crops to a wide array of crops. The key objectives of these studies, in addition to some of them comparing the effect of different farm management practices, were focused on evaluating which crops have the highest *NWP* values and are, therefore, best suited to be grown in the study regions.

Chibarabada et al. [68] and Nyathi et al. [62] benchmarked the *NWP* of traditional vegetables to alien (imported) vegetables in sub-Saharan Africa, while Nyathi et al. [59] and Nyathi [61] compared the *NWP* of traditional vegetables (such as amaranth, spider flower and orange fleshed sweet potato) to a reference (commercialized) crop (Swiss chard) in South Africa. Nyathi et al. [62] assessed that the traditional vegetables as compared to the alien (imported) vegetables were comparable in terms of their *WP*, but superior in terms of their *NWP* values for micronutrients (iron and zinc content). Nyathi et al. [59] and Nyathi [61] also assessed that the *NWP* values (for iron and zinc) varied between the traditional vegetables (amaranth, spider flower, orange fleshed sweet potato and Swiss chard) and different agricultural inputs (water and fertilizer) applications. Nyathi et al. [59,62] and Nyathi [61] demonstrated the potential of *NWP* as a very useful indicator to help assess resource (water and fertilizer) use efficiency in terms of nutritional value produced by different crop and their cultivation conditions.

The studies by Wenhold et al. [24] and Mabhaudhi et al. [21] cited the *NWP* values from Renault and Wallender [25], Mdemu et al. [30] and Brooks and Grauenhorst [32] for different animal and crop products. Wenhold et al. [24] further categorized and compared the *NWP* values of different food groups. On the other hand, several studies (see Table A1 in Appendix A) assessed *NWP* associated with different diets.

As stated before, Renault and Wallender [25] first used the *NWP* method to compare the water productivity in terms of nutritional value between different diets with varying degrees of animal products. Malmquist [29] examined the *NWP* for three different diets varying in income level using food production with examples from Ethiopia, Tanzania and Burkina Faso. Furthermore, Blas et al. [52] compared the current Spanish diet to the recommended Mediterranean diet. Mirzaie-Nodoushan et al. [58] compared the current Iranian diet to different optimized diets regarding water footprints and health. For the classification in the respective diets, the *NWP* for the different food products or food groups consumed in the diets was calculated. A robust quantification of the *NWP* of different food products offers a building block of key information to help assess and optimise water footprints with healthy nutritious diets [25].

*3.2. Methodology Approaches of NWP Studies*

3.2.1. Included Water Flows

A quantification of *NWP* value requires accounting of water flows in the agricultural production system analysed (Equation (1)). Interestingly, the review identified clear differences in the water flows included or excluded in the calculation of *NWP* in different studies. Most of the studies could be divided into three water flow categories, following from Drastig et al. [4], (1) accounting evapotranspiration water stemming from precipitation, (2) evapotranspiration water stemming from technical water, and/or (3) inclusion of all of the withdrawn technical water (Figure 3).

Most of the studies considered both the evapotranspiration from precipitation and technical water in their assessment of *NWP* values (Figure 3). Only six studies were solely concerned with evapotranspired water stemming from precipitation. Seven studies focused solely on the effects of technical water (Figure 3). Only two of these seven studies included all of the withdrawn technical water, the other studies included the evapotranspiration (water consumption) of technical water.

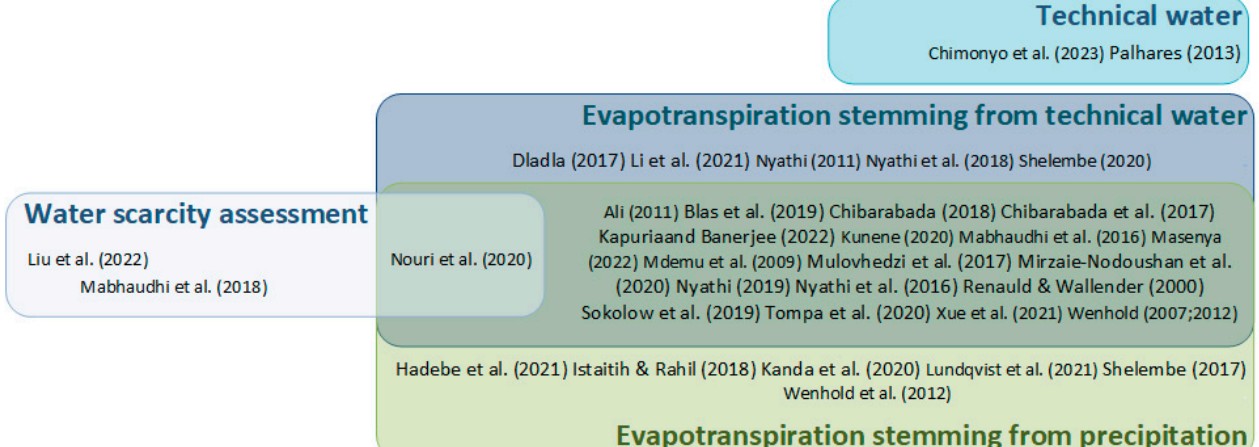

**Figure 3.** An analysis of water flow and environmental impact assessment in the nutrient water productivity studies. Studies included here: Ali (2011) [51]; Blas et al. (2019) [52]; Chibarabada et al. (2017) [53]; Chibarabada (2018) [54]; Chimonyo et al. (2023) [55]; Dladla (2017) [23]; Hadebe et al. (2021) [40]; Istaitih and Rahil (2018) [27]; Kanda et al. (2020) [46]; Kapuria and Banerjee (2022) [41]; Kunene (2020) [56]; Li et al. (2021) [50]; Liu et al. (2022) [57]; Lundqvist et al. (2021) [28]; Mabhaudhi et al. (2016) [21]; Mabhaudhi et al. (2018) [44]; Masenya (2022) [47]; Mdemu et al. (2009) [30]; Mirzaie-Nodoushan et al. (2020) [58]; Mulovhedzi (2017) [42]; Nouri et al. (2020) [3]; Nyathi (2011) [48]; Nyathi et al. (2016) [59]; Nyathi et al. (2018) [60]; Nyathi (2019) [61]; Palhares (2013) [63]; Renault and Wallender (2000) [25]; Shelembe (2017) [45]; Shelembe (2020) [64]; Sokolow et al. (2019) [31]; Tompa et al. (2020) [65]; Wenhold et al. (2007) [66]; Wenhold et al. (2012) [24]; Xue et al. (2021) [67].

Whilst most of the reviewed *NWP* studies indicated the origin of the water flows, Nyathi [43] and Malmquist [29] used databases and values from various sources without addressing the origin of the water included. Also, water quality aspects in relation to *NWP* were not discussed in any of the reviewed *NWP* studies, except the study by Blas et al. [52]. They did include water quality aspects in their discussion of the water footprint and *NWP* associated with a Mediterranean diet and food consumption patterns in Spain.

Another aspect is a robust consideration of the environmental impacts of water consumed in different agriculture production systems [18]. However, an assessment of water scarcity impact was only addressed in three publications; however, it was not directly discussed relating to *NWP* (Figure 3).

Recommendations: A sound assessment of water use in agriculture should apply a metric or set of metrics for evaluating water and nutrition productivity, as these are valuable for a water-food-nutrition-health nexus approach, and for the mitigation of local environmental impacts in terms of water scarcity and water quality. Application of *NWP* is promoted to optimize a sustainable use of limited water resources, while maximizing nutrient content of crop yields and improving food and nutrition security. This is especially important in regions where water resources are scarce or under stress. However, existing applications of NWP are very limited in terms of accounting for water scarcity impacts associated with the production of nutrient values by agriculture production systems across different regions. A water scarcity assessment provides insights into the potential water impacts associated with a production system. The FAO LEAP Water TAG guidelines [18] recommended the two methods, AWARE [17] and BWSI [8], for the assessment of water scarcity impact of livestock production systems and supply chains. Van Noordwijk et al. [69] proposed a set of hydrological indicators at the watershed level that represents the individual processes and environmental impacts of different watershed functions. It is therefore recommended to further develop and include the assessment of the potential environmental impacts associated with water use, e.g., water scarcity impact (*WSI*) [10] associated with *NWP* values of different agriculture production systems. As per the FAO LEAP Water TAG guidelines [18], the values of *NWP* and *WSI* could be integrated in a

matrix of low, medium and high *NWP* and *WSI* to help assess the urgency to act (relative environmental impact) and scope for resource use improvement (relative nutritional water productivity) of agricultural production systems. The combined metrics from these two methodologies, *NWP* and *WSI*, could provide an understanding of the pressure exerted by agriculture production sectors on water resources worldwide to support a potential improvement in nutrient water productivity as well as a reduction in its contribution to water scarcity. This is in line with the Sustainable Development Goals (SDG), where the target of SDG 6.4 deals with water scarcity. The recently published SDG 6 synthesis report presents the status of global water scarcity [70]. SDG-Target 6.4 states "By 2030, substantially increase water-use efficiency across all sectors and ensure sustainable withdrawals and supply of freshwater to address water scarcity and substantially reduce the number of people suffering from water scarcity". As such, one of the goals of a sound *NWP* assessment should be to address potential risks of water scarcity and identify key processes for improvement through the assessments of *NWP* and water scarcity impact (*WSI*) of agricultural production systems.

### 3.2.2. Accounting of Nutritional Value

Figure 4 summarizes the nutrients analysed in the existing *NWP* studies. Not included in Figure 4 are two studies which analysed a nutrient density standard for fruits and vegetables as nutrient density score (NDS) [31], as nutritional value per 100 g of the product [65]. Also not shown in Figure 4 are the publication by Li et al. [50], which is the only one that reported values for phytochemicals and total antioxidant activity, Kanda et al. [46] which reported values for ash, and Hadebe et al. [40] which reported values for starch.

Most of the studies reported *NWP* values for both some macronutrients (energy, protein, fat and carbohydrates) and some micronutrients (minerals and vitamins) (Figure 4). Three of the studies, Lundqvist et al. [28], Wenhold et al. [24], and Blas et al. [52] had a very extensive selection of nutrients and reported on all the macronutrients and a wide array of minerals and vitamins. Concerning the macronutrients, a few studies considered values for carbohydrates (four studies) and fibre (seven studies), while most of the studies considered protein, energy, and fat as macronutrients (Figure 4). Xue et al. [67] described how, according to the [37], three macronutrients (protein, energy, and fat) should be considered in evaluating the *NWP* of agricultural production systems. Additionally, many studies considered more specific values for certain minerals (25 studies) and vitamins (16 studies) (Figure 4).

Most of the studies selected the nutrients according to local nutrient deficiencies. As many of the publications come from South Africa, for example, numerous studies included the *NWP* values for Iron, Zinc and Vitamin A, as these are common deficiencies in South Africa [43,59,61]. Additionally, calcium is included in most of the studies, as it is vital for preventing malnutrition [25]. Two studies only recounted selected information from nutritional values calculated by others, e.g., Wenhold et al. [66] and Mabhaudi et al. [21] recounted *NWP* values from Renault and Wallender [25], Brooks and Grauenhorst [32], and Mdemu et al. [30].

Recommendations: A robust accounting of nutritional value is paramount in quantification of *NWP* values (Equation (1)) and optimisation of the use of limited water resources for improving food and nutrition security. However, a range of macronutrients (protein, fat and carbohydrates) and micronutrients (vitamins and minerals) are required to a target dietary recommendation. Furthermore, there is point of difference between nutrient content and nutritional value, considering variable bioavailability of different nutrients from different food items. This makes it challenging to account for different nutrient contents produced by an agriculture production system into a single nutritional value as a nominator in quantification of *NWP* (Equation (1)).

**Figure 4.** An overview of different nutrients assessed by the 40 existing nutrient water productivity studies of different food groups and primary derivatives thereof and feed. Food groups and feed chosen in the figure oriented towards the classification in Destatis [33–35], EFSA [36], FAO/INFOODS [37] and USDA [38]. Studies included here: Ali (2011) [51]; Blas et al. (2019) [52]; Chibarabada et al. (2017) [53]; Chibarabada (2018) [54]; Chimonyo et al. (2023) [55]; Dladla (2017) [23]; Hadebe et al. (2017) [39]; Hadebe et al. (2021) [40]; Istaitih and Rahil (2018) [27]; Kanda et al. (2020) [46]; Kapuria and Banerjee (2022) [41]; Kunene (2020) [56]; Li et al. (2021) [50]; Liu et al. (2022) [57]; Lundqvist et al. (2021) [28]; Mabhaudhi et al. (2016) [21]; Mabhaudhi et al. (2018) [44]; Malmquist (2018) [29]; Masenya (2022) [47]; Mdemu et al. (2009) [30]; Mirzaie-Nodoushan et al. (2020) [58]; Mulovhedzi (2017) [42]; Nouri et al. (2020) [3]; Nyathi et al. (2016) [59]; Nyathi et al. (2018) [60]; Nyathi (2011) [48]; Nyathi (2019) [61]; Nyathi et al. (2019a) [43]; Nyathi et al. (2019b) [62]; Palhares (2013) [63]; Ramputla (2019) [49]; Renault and Wallender (2000) [25]; Shelembe (2017) [45]; Shelembe (2020) [64]; Sokolow et al. (2019) [31]; Tompa et al. (2020) [65]; Wenhold et al. (2007) [66]; Wenhold et al. (2012) [24]; Xue et al. (2021) [67].

Sokolow et al. [31] used nutrient profiling, based on the recommended nutrient intakes (RNI) established by World Health Organization and FAO [71] for males between 19 and 65 years old, to quantify nutrient density score (*NDS*) as the mean of percent *RNI* per 100 kcal (energy density) of a crop. Refer to Sokolow et al. [31] for more details on calculations of *NDS* values for a crop. Sokolow et al. [31] suggested the *NDS* as an effective measure of the nutrition contribution of a crop. They compared the quantified *NDSs* of the selected crops with water footprint to highlight trade-off or synergies between nutrition produced and water consumed in crop production systems. However, further research analysis is recommended for a robust evaluation of various nutrient density indices

(such as *NDS*) to develop a standardized metric of nutrition contribution as nominator in quantification of *NWP* (Equation (1)) of agricultural production systems.

Additionally, Nyathi et al. [59–61] suggested that further research should consider nutrient bioavailability. Nutrition contribution of a food item is further influenced by different food processing and preparation methods (such as raw, cooked, blanched, boiled, fried, steamed and grilled) [56,60,61]. In particular, the factors affecting the bioavailability of nutrients with the preparation method should be explored to find out which is the best way to prepare vegetables so that nutrients have the best bioavailability for human consumption. Nyathi [48] suggested that it is also important to determine at which growth stage the crops have the highest nutritional content, and thus recommended sampling the crops per plot, at different stages of crop growth. Nyathi et al. [62] also called for further assessment of major factors that determine the nutritional content of vegetables. Their study indicated that vegetables nutritional content is variable dependent on several variables, but it is unclear which ones.

Further analysis of *NWP* of different crops, crop species and a greater range of nutrients is encouraged. Malmquist [29] suggested that more studies for specific crops and their nutritional content for representative locations and production systems are necessary to perform accurate *NWP* calculations. Furthermore, Malmquist [29] called for nutritive content of a wide range of crop varieties to be studied, as nutritional content varies within crop species. Mulovhedzi [42] also recommended future research on more crops and cultivars and an expanded range of nutrients in crop production systems in South Africa. Shelembe [64], examining the *NWP* of taro, recommended studying other taro landraces additionally to the ones explored in the study. They also recommended studying the effects of intercropping taro landraces with other crops, such as cereals, legumes and other root crops, to obtain an overview which work best in regards to *NWP*. Shelembe [45] detailed that they had to use the same nutrient content data for crops of different genera and species for their study due to a lack of genera and species-specific nutrient data. They recommended further studies to compare the genetics, morphology, physiology and nutritional value of crops within the same genera and species, in order to obtain more reliable results of *NWP* values for the specific crop species.

### 3.2.3. Assessed Farming Measures

Evaluation of the various management measures is paramount to optimize water resource efficiency in terms of nutrient value produced per unit of water consumed. A majority (23 out of the 40) studies assessed the influence of different farming measures on the *NWP* of different plant (crop) and livestock production systems (Table 2). Most of the studies assessed the effects of selection of different crops, and irrigation and fertilization regimes on the *NWP* of various crops. Additionally, the effects of variations in climate conditions and production environment on *NWP* of different crops were analysed by three studies: Chibarada et al. [53], Mdemu [30] and Xue et al. [67]. Palhares [63] assessed the effects of feeding strategies and other management practices on *NWP*, quantified in terms of kcal per litre of drinking water, for a pig production system in Brazil.

Recommendations: The existing studies highlight effects of different farm management measures on *NWP* values of agricultural production systems. However, the existing studies had assessed potential effects of farming measures on *NWP* of only few agriculture production systems and few agro-climatic conditions.

Kanda et al. [46] recommend further studies to assess the effect of different agronomic factors, such as fertilizer deficiency on cowpea's *NWP* under different irrigation regimes. Chibarabada et al. [53] also called for more experimental data and for future studies to assess the effect of different farm management practices, climate and edaphic factors on *NWP* for a range of legumes in at three sites in KwaZulu-Natal, South Africa. Nyathi et al. [62] also recommended investigating how irrigation and fertilizer affect *NWP* of a wide range of vegetables in South Africa. They also recommended a study to be carried out that evaluates the effects of leaf harvesting on yield and nutrient concentration of storage

organs, as certain vegetables could be used as dual-purpose crops. The dual-purpose crops would help to improve food and nutrition security, as this spreads food availability over a longer period.

**Table 2.** A summary of farming measures assessed by existing nutrient water productivity studies.

| Farming Branch/Processing Step | Farming Measures Covered in the Studies | Study |
|---|---|---|
| **Plant production** | | |
| selection of crops | cropping patterns, intercropping, cultivar selection | [3,23,40,42,45,55,57,67] |
| soil preparation + seedbed preparation | | |
| Sowing | planting date, planting density | [56] |
| plant protection | effect of nematodes | [49] |
| fertilization | fertilization | [3,43,56] |
| irrigation | different irrigation regimes | [3,23,42–44,50,53,55,56,59–61,64,68] |
| harvesting | harvesting, harvesting method | [43,59] |
| **Livestock production** | | |
| Feeding | feeding strategies | [63] |
| drinking, cleaning, cooling | | |

Mulovhedzi [42] recommended trials to be planted in different agro-climate regions to validate and improve *NWP* values of sweet potato (Ipomoea batatas), while Wenhold et al. [24] suggested that further research should focus on gathering data for key nutrients under a range of production conditions in South Africa. Dladla [23] also suggested future studies determine the *NWP* of sweet potato, a crop they studied under controlled conditions, under field conditions and different agro-ecologies in in a growth tunnel at the University of KwaZulu-Natal's Controlled Environment Facility in South Africa.

Nyathi et al. [60], who studied traditional South African leafy vegetables, proposed further research to assess *NWP* of traditional leafy vegetables and alien vegetables to be conducted in different locations, including different soil types and climates, and under various agronomic management practices. These included different levels of fertilizer and water stress, plant density and planting date. Nyathi et al. [59] recommended further research exploring the effects of soil fertility and water levels on the nutritional content of selected crops in different locations under various external conditions.

Additionally, they suggested that further research focus on improved varieties and cultivation practices for increased nutrition content and yield of traditional vegetable crops to South Africa. Shelembe [64] recommended to consider the effects of different planting environment and agronomy management practices such as planting density on Taro [*Colocasia esculenta* (L.) Schott].

### 3.3. Origin and Quantification of NWP Values

The quantification of *NWP* (Equation (1)) requires measures of nutrition contribution and water consumed of an agricultural production system. Out of the 40 studies reviewed, 13 calculated *NWP* values based on the nutrient content and water data values of different food groups and primary derivatives thereof and feed taken from the literature and databases (Figure 5). Two studies modelled the crop water consumed (*ET*) using the software CROPWAT version 5.7 [72] and took data on crop yield/nutrient content from databases [25,41]. Five studies modelled *ET* and crop yield and took nutrient content from a database [3,48,56,57,67]. The models used were AquaCrop [3,48], SWAT [67] and APSIM [56].

**Figure 5.** An overview of different methods applied by existing nutrient water productivity studies at different food groups and primary derivatives thereof and feed. The two main NWP-derivation methods shown here are NWP calculations or values based on literature studies. Food groups and feed chosen in the figure oriented towards the classification in Destatis; EFSA; FAO/INFOODS; and USDA [33–38]. ET: evapotranspiration [$m^3$/ha], yield: actual harvested yield [kg/ha], NC: nutrition content per kg of product [nutrition unit/kg]. Studies included here: Blas et al. (2019) [52]; Dladla (2017) [23]; Hadebe et al. (2017) [39]; Hadebe et al. (2021) [40]; Istaitih and Rahil (2018) Kapuria and Banerjee (2022) [41]; Kunene (2020) [56]; Li et al. (2021) [50]; Lundqvist et al. (2021) [28]; Mabhaudhi et al. (2016) [21]; Mabhaudhi et al. (2018) [44]; Malmquist (2018) [29]; Masenya (2022) [47]; Mdemu et al. (2009) [30]; Mirzaie-Nodoushan et al. (2020) [58]; Mulovhedzi (2017) [42]; Nouri et al. (2020) [3]; Nyathi (2011) [48]; Nyathi et al. (2016) [59]; Nyathi et al. (2018) [60]; Nyathi (2019) [61]; Nyathi et al. (2019a) [43]; Nyathi et al. (2019b) [62]; Palhares (2013) [63]; Renault and Wallender (2000) [25]; Shelembe (2017) [45]; Sokolow et al. (2019) [31]; Wenhold et al. (2007) [66]; Wenhold et al. (2012) [24]; Xue et al. (2021) [67].

Interestingly, 18 of the studies were field trials or experiments. Of these, two studies classified themselves as experiments, one under controlled conditions (without further specifications) [23] and other under semi-controlled conditions in South Africa [64]. One study was specified as a rain sheltered field experiment in South Africa [60], and other as a non-heated solar greenhouse experiment in the arid area of Northwest China [50]. Three studies were described as mixed on station and farm experiments [48,53,59] in South Africa. Ali [51] did not reveal any information about the literature/sources or modelling process of water flows, crop yield and data base of the nutrient content of the various included food groups of the study.

### 3.3.1. Examples of NWP-Values

Figure 6a–d reproduce exemplary *NWP* values of the existing studies analysing energy output per water input ($NWP_{Energy}$, kcal/$m^3$) for four food groups; (a) livestock products: meat, eggs, milk and butter; (b) cereal grains and similar: rice, maize, wheat, sorghum

and barley; (c) starchy roots and tubers: potatoes and sweet potato; and (d) legume seeds: pulses (excluding soybeans), soybeans, groundnuts and peas. It is important to note that the *NWP* results reported in Figure 6 cannot be directly compared, as the quantification of nutrition produced and water consumed are not standardized across the studies. The quantified *NWP* values reported the energy productivity of livestock products was low (between 85 kcal/m$^3$ for beef to 647 kcal/m$^3$ for milk), and conversely, cereals are high (from 1428 kcal/m$^3$ for sorghum to 2967 kcal/m$^3$ for maize). Legume seeds are also assessed for producing high energy (from 870 kcal/m$^3$ for pulses to 2060 kcal/m$^3$ for soybeans) per unit of water used. Starchy roots and tubers were quantified as the most productive group, with the sweet potatoes at 5481 kcal/m$^3$ and potatoes at 5727 kcal/m$^3$.

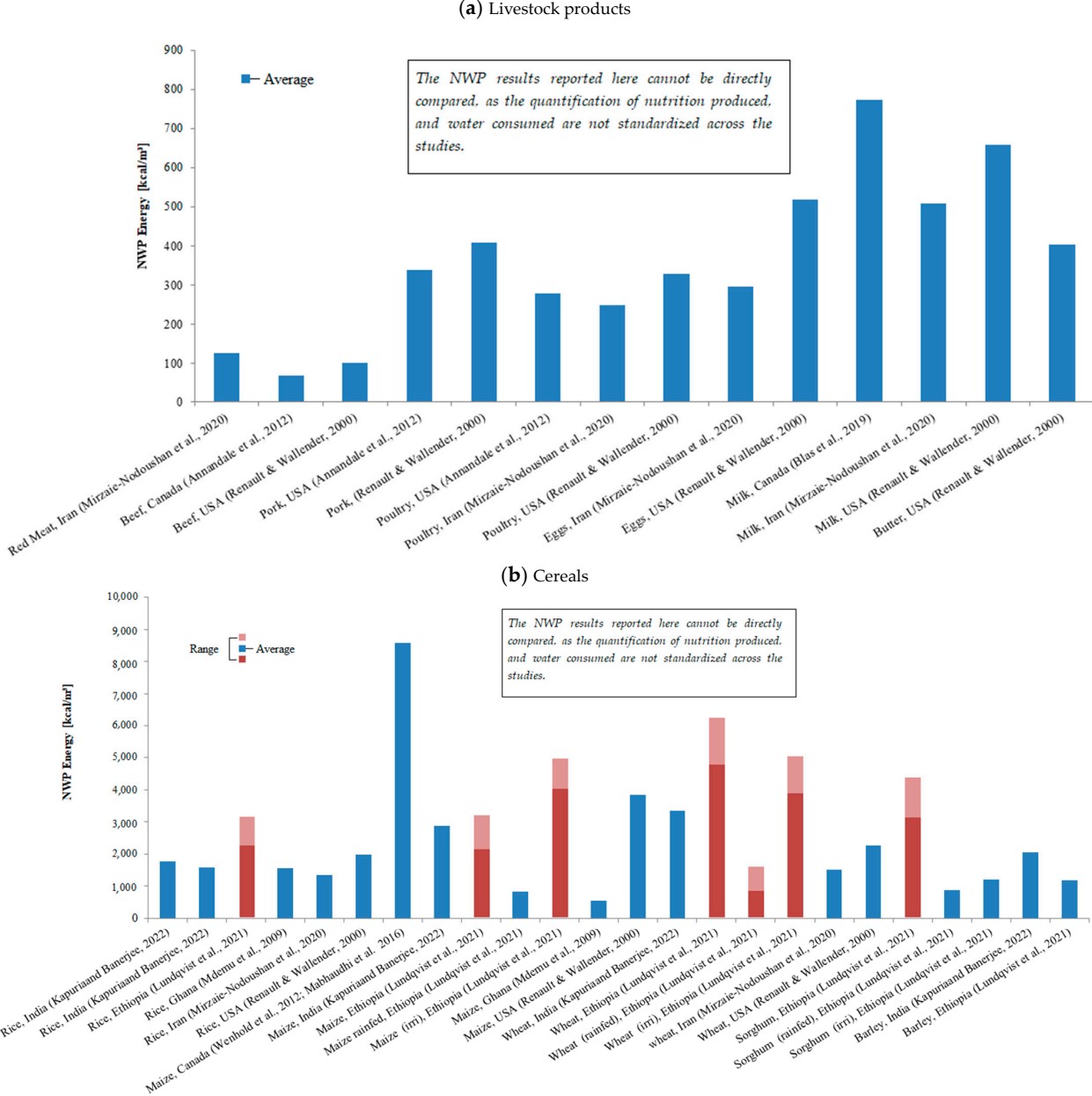

**Figure 6.** *Cont.*

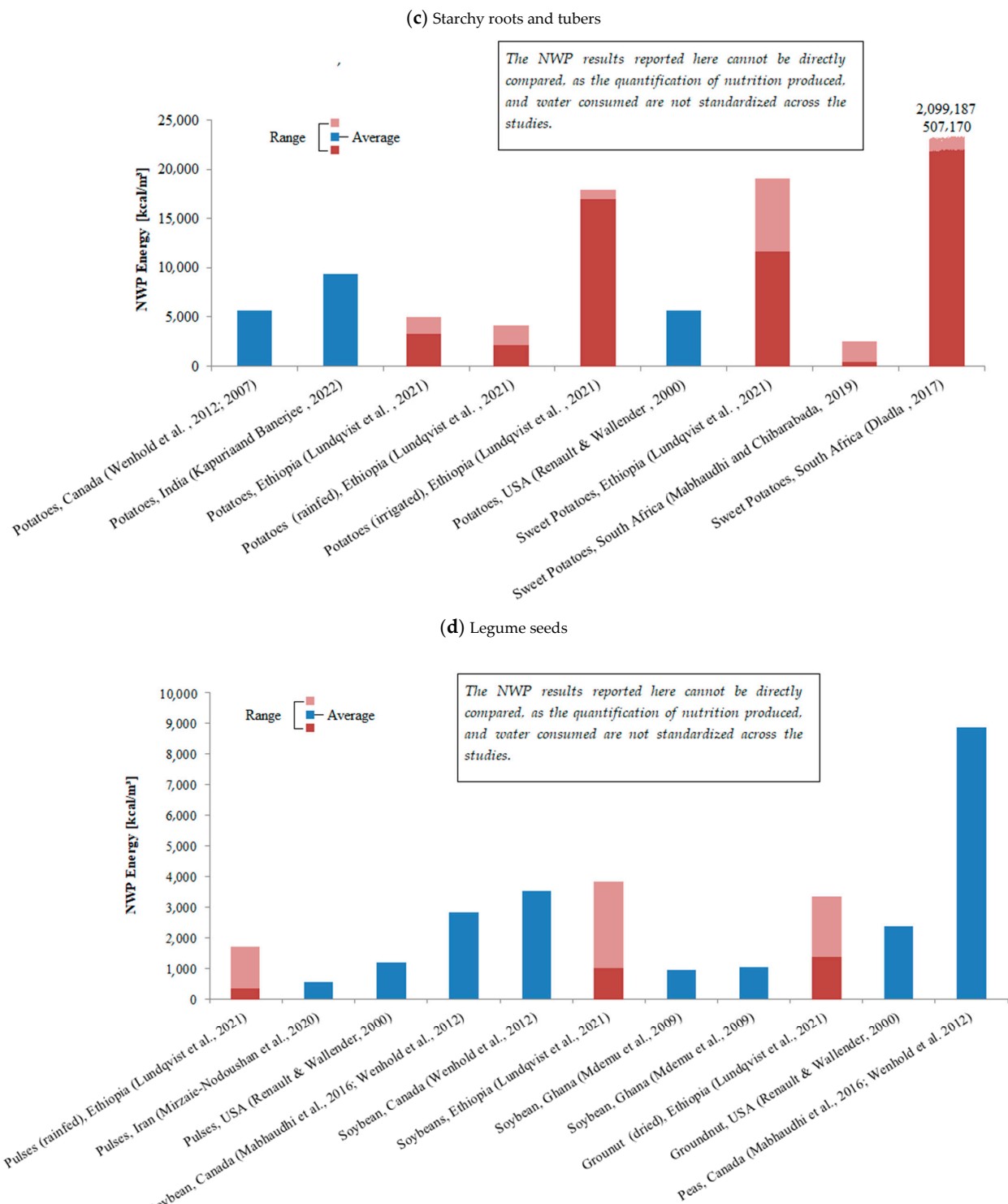

**Figure 6.** Examples of nutrient water productivity (*NWP*) values quantified on a food energy (kcal) base per unit of water used for (**a**) livestock products: meat, eggs, milk and butter, (**b**) cereal grains and similar: rice, maize, wheat, sorghum and barley, (**c**) starchy roots and tubers: potatoes and sweet potato, and (**d**) legume seeds: pulses (excluding soybeans), soybeans, groundnuts and peas. Note: these NWP results cannot be directly compared, as the quantification of nutrition produced, and water consumed are not standardized across the studies. Studies included here: Blas et al. (2019) [52]; Dladla (2017) [23]; Kapuria and Banerjee (2022) [41]; Lundqvist et al. (2021) [28]; Mabhaudhi et al. (2016) [21]; Mabhaudhi et al. (2018) [44]; Mdemu et al. (2009) [30]; Mirzaie-Nodoushan et al. (2020) [58]; Renault and Wallender (2000) [25]; Wenhold et al. (2007) [66]; Wenhold et al. (2012) [24]; Xue et al. (2021) [67].

However, the methodological differences (e.g., the inclusion of evapotranspiration water stemming from precipitation stored as soil moisture 'green water' and evapotranspiration water stemming from technical 'blue' water used for irrigation and the inclusion of all of the withdrawn technical water in water input) and energy produced (*kcal*) limit the informative value of the reported values (Figure 6) for a relative comparison of the *NWP* of different food groups. Also, various studies noted their limitations, for example plant damages due to animal attacks or weather conditions in trials [53,64], untypically low water productivity values attributed to low crop yield due to management practices [30], calculation errors [56], yield data handling, disease and pests and topography effects [48].

Moreover, the origin of input data used in the calculation of *NWP*-values based on field trials, modelling or taken from existing databases has potentially inherent different uncertainties in a robust quantification of *NWP* of different food groups. The uncertainty of *NWP* calculations based on different values is expected to vary as follows:

Lower Uncertainty

water consumed (*ET*), yield, and nutrient content based on field measurements
*ET* and yield based on field trials, nutrient content from existing databases
*ET* modelled, yield based on field trials, nutrient content from existing databases
*ET* and yield modelled, nutrient content from existing databases
*ET* modelled, yield and nutrient content from existing databases
*ET*, yield, and nutrient content from existing databases

Higher Uncertainty

### 3.3.2. Lack of Reliable Data

A lack of available quality input data impacts the reliability of the *NWP* results. According to Lundqvist et al. [28], access to and quality of reliable data is currently a basic challenge in the calculation of *NWP* across a range of crops and other produce ranging from rainfed to irrigated agriculture in Ethiopia. Malmquist [29] also described similar problems, as values of water productivity for individual crops and food categories could not be stated due to gaps in water productivity values in rainfed agriculture in Sub-Saharan Africa. Furthermore, insufficient data for reliable calculations and reporting of *NWP* values is mentioned by several other studies. Li et al. [50] did not find experimental data on tomatoes cultivated in arid areas of Northwest China. Shelembe [45] had to use crops belonging to different genera and species to complement missing data for the calculation of a legume—leafy vegetable intercrop system in KwaZulu-Natal South Africa.

As stated by Wenhold et al. [66], water consumption and all of its related metrics, such as crop yield, depend on different factors, such as the geographical location, climate and the duration of the cropping season. Therefore, when information on the water use of a particular crop is obtained at a specific site, it cannot necessarily be used for further calculations of the same crop at a different site. Renault and Wallender [25] also mentioned that the values of water productivity and yield recorded in California are likely higher than in some other developed countries and thus not appropriate for other countries or general global studies. Blas et al. [52] also pointed out that the data they used for their water productivity calculations showed a wide range of estimations for the same products and countries. They attributed this to local climate variabilities and local agricultural practices having a significant effect on water use.

However, due to the scarce available information, *NWP* values are often calculated for a particular location using data from a different site. Mabhaudhi et al. [21] and Lundqvist et al. [28] pointed out that their studies had to use values from separate studies and databases, and thus, different locations and experimental designs made the results less reliable. For example, Blas et al. [52] mentioned the multiple datasets they had to use for their study. They highlighted how difficult it was to obtain reliable data and that they could only analyse one-year data, which clearly limited their results. Renault and Wallender [25]

also described how they had to use different datasets for the nutrition content of the product, which showed significant deviations. They illustrated this vast difference by giving an example of the energy content of cereals in the US, which varies from 2700 kcal/kg in the FAO balance sheets (1995) to 3500 kcal/kg in the report by Dunne [73]. They explained this deviation by the differences in processing, so if the values have been recorded as a raw product, as partially processed or after cooking. However, most of the values reported in the literature do not specify the level of processing. This lack of transparency is another factor impacting the reliability of *NWP* results, given that it is often not possible to trace back where the deviations come from.

Wenhold et al. [24] could not find enough data to make the calculations for the selected food products. Thus, they made estimates on the *NWP* of the selected products using two independent data sources from separate trials, a crop water productivity database and a nutritional content database, which both showed some uncertainty. The parameters used for their estimates of *NWP*, such as yield or biomass, evapotranspiration and nutrient concentration, differ among crops, management practices and locations due to climatic conditions, soil fertility and water availability.

Given that the comparison for nutritional and agronomic practices is only valid when grown under the same conditions, putting the two different databases together to create a third database on *NWP* values raises concerns about the reliability of the *NWP* results obtained. Wenhold et al. [24] speculated that errors in calculating *NWP* with two independent data sets caused the wide range in *NWP* values for all the food products in their study. They stated that their approach could be better and highly encouraged these *NWP* calculations to be made with single reliable data sets. However, it was the only available approach for the first-order estimates of *NWP* values given the limited information.

Nyathi et al. [43], who aimed to benchmark the *NWP* values of twenty vegetables, detailed the same limitations as Wenhold et al. [24]. Nyathi et al. [43] also combined a crop productivity dataset and a nutrient concentration database to create a third *NWP* database. Moreover, Li et al. [50] mentioned that information on available crop nutritional yield and *NWP* were minimal and that there were no experimental studies they could have used in calculation of *NWP* values of for tomatoes grown in a greenhouse in the arid area of Northwest China. Furthermore, adding to the issue of a lack of transparency, Li et al. [50] reported that for the water use values to that were used to generate the water productivity database, which was sourced from the literature, some studies did not indicate whether water use was total water applied or evapotranspiration.

Recommendations: Research efforts are required to develop further reliable databases and models for robust quantification of *NWP* of different agricultural production systems under different agroclimatic conditions. However, long-term field trials are practically challenging to quantify water consumed, yields and nutrient content of different agricultural production systems. A robust integration of field observations with locally calibrated agrohydrological models offers the potential to develop reliable datasets further to quantify the *NWP* values of agricultural production systems.

Agrohydrological models such as the AquaCrop model predict the water use and crop yield data and metrics necessary for *NWP* calculations. Nyathi [48] suggested that stakeholders such as district managers, extension officers and farmers use the AquaCrop model for deciding on the different combinations of the levels of water stress in relation to growing spinach in Roodeplaat, Republic of South Africa. The AquaCrop model was recommended to be further calibrated and validated in more fields to see how well it predicts crop yield and water use under different production environments. Further recommendations are directed at making the results of their study more generic with model calibration. Here, Nyathi et al. [43] recommended that crop growth models such as AquaCrop, the soil water balance (SWB) model, the agricultural production system simulator (APSIM), and the world food studies (WOFOST) model be calibrated and validated by using additional new data generated from the same experiment for aboveground biomass, storage root yield,

evapotranspiration and water productivity. This would help to upscale their results to other locations.

Hadebe et al. [40] also suggested that the data from field trials be additionally used to extend existing crop water productivity models to include nutritional water productivity modelling for numerous crops and genotypes. Kunene [56] suggested that factors affecting the growth and productivity of African leafy vegetables, the object of their study, be further studied to provide new bases for improved *NWP* modelling. Analysing these factors would help comprehend the effects of different production environments on the concentration of nutrients and, therefore, minimize the challenges faced when modelling nutritional water productivity.

Furthermore, Nyathi et al. [60] encouraged using geographical information Systems and remote sensing to scale up *NWP* results throughout South Africa. Nyathi et al. [43] recommended creating reliable, steady new variables, such as for nutritional yield, water productivity and *NWP*, using datasets collected from the field experiments, as due to a lack of reliable datasets, they had to use data from different sources, which limits *NWP* reliability.

Wenhold et al. [24] recommended future research to work with single reliable datasets, as working with independent datasets due to a lack of available data likely caused errors in their calculations. This calls for a robust integration of field measurements, locally calibrated and validated agrohydrological and farm production models, and GIS and remote sensing tools to quantify reliable *NWP* values for different agriculture production systems under different production environments. This is expected to support consistency and reliability in applications of *NWP* assessment to optimise the use of limited water resources for improving food and nutrition security.

### 3.3.3. Implementation of NWP Findings

A robust *NWP* assessment aims to help optimise use of local resources for improved food and nutrition outcomes. However, a further challenge for *NWP* assessment is consideration of the local context. According to Lundqvist et al. [28] a concern regarding *NWP* values of animal products appears to be their comparison with *NWP* values of plant products without considering the local context. Animal products could be from cattle grazing in areas where there are no other realistic options to use the available land if required to be used for food production. Similarly, when looking at crops that are used for livestock feed, mainstream calculations often dismiss that these may be produced in environments that are not suitable to grow crops which are intended for human consumption.

A portfolio of suitable crop and livestock products is expected to match nutritional requirements, rather than a crop specific as no single crop could meet all nutritional demands. Therefore, a robust assessment of *NWP* should allow for a lifecycle assessment of the components of the on-farm crop portfolio (rather than a single crop), clearly put in context of local production environment and water scarcity impacts at the relevant catchment scale.

Lundqvist et al. [28] also mentioned the ability of farmers to adapt their production system to the recommended crops for more nutritionally valuable and effective production. Farmers could face challenges to change their production system, as it takes time and effort to learn how to grow the new crops efficiently and economically rewarding. On top of that, growing certain crops can represent an additional risk to farmers if they are more vulnerable in their production due to lack of required inputs such as fertilizer and irrigation water, and pest disease attacks. For example, high-value crops such as fruits and vegetables are more sensitive to moisture stress and many high-value crops which are dense in important nutrients such as fruits, vegetables and leguminous crops require proper post-harvest arrangements [28].

Additionally, the demand and prices for nutritionally more efficient crops can often be volatile [28]. Considering all these factors, it is clear that farmers take a huge risk when adjusting to new agriculture production conditions and that these conditions can

represent an immense drop in revenue for them. Therefore, they are unlikely to take a risk of abandoning their steady and less risky cultivation in favour of highly nutritious ones, even if they might be more economically promising in the long run [28]. Nouri et al. [3] also mentioned the market-oriented nature of the crop pattern selection in the Upper Litani Basin, where the lower profitability of crops could make it difficult to transition towards more nutritionally valuable and water-efficient production.

This calls for a robust policy and planning support for a robust implementation of *NWP* findings to optimise the use of limited water resources for improving food and nutrition security.

## 4. Concluding Remarks

This review of existing studies demonstrates that significant research efforts are being dedicated to conceptualising and developing of methods, tools and databases for assessing *NWP* of agricultural production systems. The concept of *NWP* is potentially as a very useful indicator to help optimise water resource use efficiency in terms of nutritional value produced in agriculture. It offers opportunities to robustly analyse and benchmark agriculture production systems, assess potential effects of farming measures, inform choice of resource efficient farming systems, and provide insights for achieving sustainable and nutritious, health diets for improved human and environmental health outcomes. A robust *NWP* assessment offers potential policy relevance in balancing human nutritional requirements and available land and water resources, by informing choice of a portfolio of suitable crops and livestock products to produce the nutrition required, the intensity of farm management (use of water, fertilizer and other inputs) to achieve efficient production systems, local-vs-import of food production and potential environmental impacts associated with different dietary choices.

However, our review highlights lack of a consistent approach and reliable data and tools supporting a robust and coordinated application of *NWP* analysis of agricultural production systems worldwide. The reviewed studies varied widely in terms of their methodological approaches, data and tools used and the water flow and nutrient content accounted in their *NWP* analysis. The studies could be divided into three water flow-accounting categories: evapotranspiration water stemming from precipitation, evapotranspiration water stemming from technical water and/or inclusion of the withdrawn technical water. However, the methodological differences in terms of accounting different water flows and nutrient contents, the scale of analysis and use of unreliable databases and tools appear to severely limit the informative value of reported *NWP* values for a robust assessment and a relative comparison of different agricultural production systems analysed. Moreover, only four studies addressed water scarcity impacts associated with *NWP* in their study region.

A robust assessment of nutritional water productivity must address not only nutritional value produced per unit of water used, but also potential risks of associated water scarcity and identify key processes for improvements of *NWP* and reduce the water scarcity impact (*WSI*) of agricultural production systems. Therefore, further development of *NWP* analysis framework in agriculture should integrate the assessment of *NWP* and *WSI* (following the ISO 14046 standard [10] and the UN's SDG 6) with a focus on potential improvements in the agriculture sector's nutritional water productivity as well as a reduction in its contribution to water scarcity. The review calls for a multidisciplinary research effort to further develop standardized metrics and appropriate scale of a robust *NWP* assessment, clearly put in context of local production environment and water scarcity impacts at the relevant catchment scale. Multidisciplinary research efforts, including soil scientists, agronomists, food scientists and irrigation and catchment hydrologists, are required to support the development of the standardized and combined metrics of *NWP* and *WSI*, and a robust integration of field measurements, locally calibrated and validated agrohydrological and farm production models, and GIS and remote sensing tools to quantify reliable *NWP* values and their associated *WSI* of agriculture production systems. This is

expected to support consistency and reliability in applications of nutritional water productivity assessments to optimise the use of limited water resources for improving food and nutrition security.

**Author Contributions:** Conceptualization, R.S. and K.D.; methodology, R.S. and K.D.; formal analysis, F.-M.T., J.J., S.Z.C. and K.D.; data curation, F.-M.T., J.J., S.Z.C. and K.D.; writing—original draft preparation, F.-M.T., J.J., R.S. and K.D.; writing—review and editing, R.S. and K.D.; visualization, K.D.; supervision, R.S. and K.D.; project administration, R.S. and K.D. All authors have read and agreed to the published version of the manuscript.

**Funding:** This research received no external funding.

**Data Availability Statement:** Not applicable.

**Acknowledgments:** The Workshop was sponsored by the OECD Co-operative Research Programme: Sustainable Agricultural and Food Systems, whose financial support made it possible for some of the invited speakers to participate in the Workshop.

**Conflicts of Interest:** The authors declare no conflict of interest.

## Appendix A

**Table A1.** A summary of farming measures assessed by existing nutrient water productivity studies. Studies are classified by studying diets and dietary components.

| Publication | Diets | Dietary Components |
|---|---|---|
| Ali [51] | Diet 0 (reference) 178 kg vegetables, 121 kg of fruits, 113 kg of cereals, 67 kg of sugar products 277 kg of milk, egg and butter, 29 kg of fat/oil and 117 kg of meat.<br>Diet 1 animal products reduced by 25% and replaced by vegetable products<br>Diet 2 50% beef replaced by poultry together with an adjustment of vegetables<br>Diet 3 50% meat replaced by vegetable products<br>Diet 4 animal products reduced by 50% and replaced by vegetable products<br>Diet 5 vegetarian<br>Diet 6 survival diet. It is based only on the four most productive products namely potato, groundnut, onion and carrot | Food groups: vegetables, fruits, cereals, sugar products, milk, egg and butter, fat/oil, meat. |
| Blas et al. [52] | two diets:<br>current Spanish and recommended Mediterranean | Food groups: fruits and vegetables; cereals, olive oil and healthy drinks; olives, nuts, seeds and condiments; dairy products; eggs and legumes; fish and seafood; potatoes; white meat and vegetable fats; red or processed meat and sugar, sweets, sauces and beverages. |
| Chibarabada [54] | two diets with grain legumes | The study benchmarked underutilized grain and cowpea to major grain legumes (groundnut) and dry bean. |
| Kapuria [41] | diet with grain production | replacing summer crop (Boro rice) in each district with maize. |
| Malmquist [29] | two diets varied in income level:<br>low-income diet, middle income diet, high income diet | Food groups:<br>Cereals, roots and tubers, pulses and legumes, oil crops, vegetables, fruits and animal products. |

**Table A1.** *Cont.*

| Publication | Diets | Dietary Components |
| --- | --- | --- |
| Mirzaie-Nodoushan et al. [58] | four diets:<br>current (reference) Iranian, healthy recommended diet under national food-based dietary guidelines and two optimized diets (minimized total consumption WF diet and minimized internal blue WF diet) | Agricultural goods from the Iranian food basket, divided into 12 groups:<br>wheat, rice, red meat, poultry, fish, milk eggs, fruits, vegetables, pulses, vegetable oil and sugar. |
| Nyathi [61] | two diets:<br>traditional vegetables and alien vegetables | Traditional vegetables: amaranth, blackjack, kale, Chinese cabbage, spider flower, jute mallow, pumpkin leaves, sweet potato leaves, black night shade, cowpea leaves.<br>Alien vegetables: onion, beetroot, Swiss chard, cabbage, broccoli, cucumber, carrot, butternut, lettuce, tomato. |
| Palhares [63] | five diets for pigs:<br>T1—high crude protein level<br>T2—high crude protein level with reduced crude protein level<br>T3—high crude protein level, inclusion of phytase and reduction of calcium and phosphorus<br>T3—high crude protein level, inclusion of phytase and reduced calcium and phosphorus contents<br>T4—high crude protein level with the supplementation of 40% organic minerals and 50% inorganic minerals<br>T5—high crude protein level but combining treatments T2, T3 and T4 | Protein level, crude protein, phytase, calcium, phosphorus, supplementation of 40% organic minerals and 50% inorganic minerals. |
| Renault and Wallender [25] | six diets varied in meat consumption: reference USA, 25% reduction animal product, poultry replace 50% beef, Vegetal product replace 50% red meat, 50% reduction animal product, vegetarian and survival | 7 animal products: bovine meat, pork meat, pork meat, poultry meat, eggs, milk and butter.<br>21 vegetal products: wheat, rice, maize, potatoes, sugar beet, pulses (beans), tree nut, groundnut, soybean oil, cotton seed oil, tomatoes, onions, orange, lemon, grapefruit, banana, apple, pineapple, dates and grape. |
| Sokolow et al. [31] | method to measure and compare the water footprint of crops in 5 food groups relative to their potential nutrient contribution to the human diet. | 17 grains, roots and tubers, 9 pulses, 10 nuts and seeds, 17 vegetables, 27 fruits. The selected crops were chosen based on the availability of the water footprint benchmark. |
| Tompa et al. [65] | Hungarian diet | 44 food items in four classifications: plant-based foods, animal-based foods, including riboflavin, including vitamin C. |

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
