# Peer review of "A Review of Nutritional Water Productivity (NWP) in Agriculture: Why It Is Promoted and How It Is Assessed?†"

_water, doi:10.3390/w15244278_

Round 1

Reviewer 1 Report

Comments and Suggestions for Authors

The NWP concept connects issues of crop choice for balanced human nutrition to the wise allocation of scarce water resources. The current manuscript has the potential to contribute conceptual clarity and operational 'devils in the details' of the way the concept has so far been used, but some further work is needed to achieve this. 

In terms of the potential policy relevance of the NWP metric the starting points may be 1) the land and water allocation issues that center on the choice of crops (in monocultures, mixed cropping and/or rotations), the intensity of management (use of irrigation, fertilizer and other inputs), 2) the requirements of balanced human nutrition and the choices in import-vs-local production, and in vegan versus omnivore human diets. 

One would expect to see a focus on the crop (and/or livestock) portfolio's that match nutritional requirements -- rather than crop-specific data as such, as no single crop can efficiently meet nutritional needs.

A major issue is that the appropriate scale of assessment for the policy choices would be one that allows for a life-cycle assessment of the components of the on-farm crop portfolio (rather than plot-level data), with clarity on local rainfall vs surface- and groundwater dependent irrigation -- at the relevant watershed scale. The WSI index addresses part of the latter issue but only partly.  

Line 16 You might start with something like: "The nutrient water productivity concept combines a metric of dry matter production per unit water use and the crop-specific human nutritional characteristics of the dry matter involved. As such it can rationalize the use of scarce water for a portfolio of crops that jointly matches human nutritional needs. For both components of the NWP concept, however, a range of operational definitions is in use in the literature, making results not directy comparable."

Line 55. A fourth class is focussed on Human impacts on water cycles (HIWC) and takes the Et of natural vegetation as site-specific reference, arguing that both additional and reduced water use can disturb landscape-level water balance (Van Noordwijk M, van Oel P, Muthuri C, Satnarain U, Sari RR, Rosero P, Githinji M, Tanika L, Best L, Comlan Assogba GG, Kimbowa G. et al. Mimicking nature to reduce agricultural impact on water cycles: A set of mimetrics. Outlook on Agriculture 2022 51(1):114-28).

Line 56 No need to start with "however", the following sentence builds on the preceding one. 

Line 59 FM needs to be defined, as it is not part of SI units

Line 61 Please clarify "It". You may need to first clarify the different scales of analysis that become involved in "indirect" water use: it is indirect from a local perspective, but maybe not when a watershed is taken as unit of analysis (depending on the sources used).

Line 70 This may be a place to clarify the potential policy relevance of including or excluding various terms of a global water balance (and the human impacts on water cycles).

Line 77 Please clarify the difference between a 'farm' versus 'fork' perspective that involves food waste issues and circularity.

Linee 90 Somewhere here one would expect a discussion of 'yield gaps' versus 'efficiency gaps' -- part of the literature suggests that efficiency gaps widen (i.e. more inputs are needed per unit production) if yield gaps are closed (say, beyond achieving 80% of local production potential). Footprint data thus refer to current levels of agricultural intensification, rather than being 'generic'.

Line 103. A further discussion of scale is relevant here: is the target a local autarchy (self-sufficiency in balanced nutrition) or an active participant in global markets where prices convey scarcity information? 

Line 104 The 2.2 is redundant -- or do you intend to have a subheading?  It would be good to state the targets of the current review around here. Maybe some of the preceding text canbe shifted to section 2, as the text in lines 116-1299 is more introductory than what precedes it.

Line 138 The Y term needs clarification where more than one crop component is harvested (e.g. grain plus stover used as animal feed). Even more so where intercropping produces multiple harvestable products. A further set of issues arises in the animal production applications where farms become a relevant scale of assessment.

Line 115 The term "promote" seems to refer to a higher-level policymaking entity than a farmer -- what are the means available to "promote"? Do different ways to promote involve different scales and hence different preferred metrics?

Line 116 "to" is redundant

Line 159 Inclusion of portuguese iss commendable, but french, spanish and chinese might yield more? I assume portuguese equivalents of the keywords were used?

Figure 6 -- Despite the disclaimer in Figure 6, showing the results carries the risk of misinterpretation. The paper would be stronger ifn these results are not shown, but only the ranges of reported values mentioned, with all the caveats that defy direct comparisons. 

Comments on the Quality of English Language

Generally OK, a few minor issues were mentioned above

Author Response

Dear reviewer,
Please find attached our point-by-point response to your commments.
Best Regards,
Katrin Drastig

Reviewer 2 Report

Comments and Suggestions for Authors

Author Response

(The authors gave the same response as above.)
